# Enhancing Patient-Centric Drug Development: Coupling Hot Melt Extrusion with Fused Deposition Modeling and Pressure-Assisted Microsyringe Additive Manufacturing Platforms with Quality by Design

**DOI:** 10.3390/pharmaceutics17010014

**Published:** 2024-12-25

**Authors:** Dinesh Nyavanandi, Preethi Mandati, Nithin Vidiyala, Prashanth Parupathi, Praveen Kolimi, Hemanth Kumar Mamidi

**Affiliations:** 1Small Molecule Drug Product Development, Cerevel Therapeutics, Cambridge, MA 02141, USA; nithinvidiyala@gmail.com; 2Department of Pharmaceutics and Drug Delivery, School of Pharmacy, The University of Mississippi, University, MS 38677, USA; preethimandati08@gmail.com (P.M.); kp08071993@gmail.com (P.K.); 3Division of Pharmaceutical Sciences, Arnold & Marie Schwartz College of Pharmacy and Health Sciences, Long Island University, Brooklyn, NY 11201, USA; reddy.parupati@gmail.com; 4Drug Product Development, Continuus Pharmaceuticals, Woburn, MA 01801, USA; mhkumar1990@gmail.com

**Keywords:** additive manufacturing, fused deposition modeling, pressure-assisted microsyringe, semi-solid extrusion, hot melt extrusion, continuous manufacturing, patient-centric, 3D printing, quality by design

## Abstract

In recent years, with the increasing patient population, the need for complex and patient-centric medications has increased enormously. Traditional manufacturing techniques such as direct blending, high shear granulation, and dry granulation can be used to develop simple solid oral medications. However, it is well known that “one size fits all” is not true for pharmaceutical medicines. Depending on the age, sex, and disease state, each patient might need a different dose, combination of medicines, and drug release pattern from the medications. By employing traditional practices, developing patient-centric medications remains challenging and unaddressed. Over the last few years, much research has been conducted exploring various additive manufacturing techniques for developing on-demand, complex, and patient-centric medications. Among all the techniques, nozzle-based additive manufacturing platforms such as pressure-assisted microsyringe (PAM) and fused deposition modeling (FDM) have been investigated thoroughly to develop various medications. Both nozzle-based techniques involve the application of thermal energy. However, PAM can also be operated under ambient conditions to process semi-solid materials. Nozzle-based techniques can also be paired with the hot melt extrusion (HME) process for establishing a continuous manufacturing platform by employing various in-line process analytical technology (PAT) tools for monitoring critical process parameters (CPPs) and critical material attributes (CMAs) for delivering safe, efficacious, and quality medications to the patient population without compromising critical quality attributes (CQAs). This review covers an in-depth discussion of various critical parameters and their influence on product quality, along with a note on the continuous manufacturing process, quality by design, and future perspectives.

## 1. Introduction

In today’s world, with the increasing patient population, the demand for tailored and complex medications is increasing enormously. However, the limitations of traditional manufacturing platforms are limiting the capability of pharmaceutical industries. Oral formulations are much preferred among various dosage forms, due to their low cost and easy administration [1,2,3,4]. Oral medications have colossal demand and generate good revenue for the pharmaceutical industry.

Knowing that “one dose fits all” is not true in all scenarios is essential. The pharmaceutical industry has researched and recommended the doses of the medications available in the market based on safety and efficacy, leaving no flexibility for physicians to adjust the dose based on the severity of disease condition, age, sex, or race [5,6,7]. Few patients might even require dosage forms with varied drug release profiles to overcome any side effects of high plasma drug concentrations. The majority of the geriatric patient population is required to be administered several medications each day depending on their disease state, which might also be challenging for patients with dysphagia [8,9,10]. Developing combination products might benefit the geriatric population to a certain extent. However, developing tailored medications to address patients’ needs by employing traditional manufacturing platforms such as direct blending, dry granulation, and wet granulation remains challenging [11,12,13,14,15].

In 2015, the approval of Spritram^®^ medication by the United States Food and Drug Administration (USFDA) developed by the Zipdose^®^ additive manufacturing platform attracted the attention of researchers from various avenues of the pharmaceutical sector [16,17,18,19]. Since then, much research has been conducted in academia and industries exploring the opportunities and applications of additive manufacturing techniques for developing tailored medications. Additive manufacturing involves fabricating a three-dimensional (3D) object by depositing the material layer by layer until the desired object is printed [14,15,20,21]. The 3D printing process is most widely employed in automobile industries for designing prototype models. It is also utilized to manufacture various complex components that are difficult to manufacture using traditional approaches [16,17,22,23,24]. Additive manufacturing provides advantages such as the on-demand manufacturing of precision medicine, dose adjustment, and the fabrication of polypills and extemporaneous clinical supplies. Even pediatric medications can be fabricated in different shapes and flavors to attract children and make them adhere to prescription medications [25,26,27,28].

To date, the various 3D printing techniques investigated for developing pharmaceutical dosage forms are classified as inkjet printing (IP), stereolithography (SLA), binder deposition, powder fusion, and nozzle-based printing techniques. The process of inkjet printing is suitable for solutions and drug suspensions. It utilizes low temperature and low pressure. The liquid material is sprayed on the build platform through a nozzle in the printer head. The deposited material requires time to cure between the printing of successive layers [27,28,29]. As the process is suitable for low-temperature additive manufacturing, the faster curing of deposited material might require infrared or ultraviolet light. The resolution of printlets developed by the IP process depends on the printer nozzle size. The SLA type of additive manufacturing involves the polymerization and solidification of resin material. Similar to the IP process, the SLA technique involves the layer-by-layer printing of material until the desired object is formed. The major disadvantage of the SLA type of 3D printing is the availability of a limited number of cross-linked polymers, which are generally considered safe (GRAS). The binder deposition type of 3D printing is a powder-based technique where a layer of liquid binder is sprayed between successive layers of powder particles. The binder deposition technique was used to manufacture the first FDA-approved commercial 3D-printed product, SPIRTAM. Another type of powder-based additive manufacturing technique is powder fusion. This approach involves the application of heat using a laser beam where powder particles are bound using a low-melting-point material [29,30,31,32]. Lastly, another type of additive manufacturing is the nozzle-based technique. This technique can be further classified as fused deposition modeling (FDM) and pressure-assisted microsyringe (PAM). FDM and PAM are thermal manufacturing techniques where the material is melted and deposited on top of the build platform until the object of desired geometry is fabricated. Though there have been various review articles published in this field, the focus of the current article is to explore the feasibility of continuous manufacturing by coupling it with the hot melt extrusion process and also to introduce quality by design for the first time within additive manufacturing. This review article discusses the critical material and process attributes essential in developing a robust dosage form utilizing nozzle-based additive manufacturing techniques, i.e., FDM and PAM. In addition, the suitability of nozzle-based additive manufacturing for establishing a continuous manufacturing line paired with the hot melt extrusion process is also discussed, along with their advantages and limitations.

## 2. Pressure-Assisted Microsyringe (PAM)

PAM, called semi-solid extrusion (SSE), is a computer-controlled extrusion process. The instrumentation resembles a “syringe and piston” assembly, as shown in Figure 1. The formulation materials loaded into the cylindrical body are extruded and deposited onto the build platform layer by layer until the object is fabricated completely [33,34]. The piston inside the barrel is guided by pneumatic or mechanical force. The other type of mechanism is a rotating screw which conveys the material towards the nozzle, and the accumulation of material generates force and extrudes the material from the nozzle. The barrel has heating flexibility or can be maintained at ambient temperature, making it suitable for processing powder and semi-solid materials. The curing time for the 3D-printed mass depends on the type of material loaded into the barrel. The powder material loaded into the barrel is melted by applying thermal energy. For such a process, the product’s curing happens readily compared to the processing of semi-solid solvent materials. The critical process parameters associated with this process include barrel temperature, build platform temperature, printing speed, and piston pressure [35,36]. Various critical material attributes, critical process, and instrument parameters that play a key role in the successful fabrication of medications are further discussed in detail.

### 2.1. Critical Material Properties

#### 2.1.1. Thermal Properties

The process of PAM or SSE involves the application of thermal energy. Thus, the thermal properties of processing materials play an essential role in successfully printing dosage forms. The thermal properties of processing materials, such as melting temperature, glass transition, and degradation temperature, need to be studied carefully. Considering the residence time of the material inside the heated barrel, the thermal process is unsuitable for heat-sensitive materials but can still be conducted by preparing a semi-solid paste using appropriate solvents. The solvent-mediated process can also be employed for materials with low degradation temperatures. For the processing of formulations consisting of a drug and polymer, while screening the materials, the glass transition temperature and melting point of the polymer must be considered, which might require high processing temperatures [37,38]. The addition of solid plasticizers such as polyethylene glycol (PEG) and stearic acid needs to be studied for processing at low temperatures. The processing temperature should be above the glass transition temperature of the processing material to avoid any clogging of materials in the nozzle and to avoid applying high extrusion pressure. It should stay below the degradation temperature of the materials. Below the glass transition temperature, the material exhibits a rigid and brittle nature, which is challenging to print. Above the glass transition temperature, the material becomes easily flowable, but its effect on the deformation of the structure needs to be monitored. Maintaining the printing temperature close to the glass transition temperature aids faster cooling, retaining structural integrity. One other aspect that needs to be studied is thermal expansion. Few thermally expanded materials shrink upon cooling, losing the bonding between layers and affecting the mechanical integrity and quality of the dosage form [39]. The processing temperature needs to be optimized by considering the drug’s desired physical state (amorphous or crystalline) within the dosage form.

#### 2.1.2. Rheological Properties

The viscosity of the processing material and the flow of material from the nozzle onto the build platform play an essential role in successfully fabricating 3D-printed medicine. Thus, it is very important to understand the rheological properties of formulation components. The optimum viscosity of the material inside the barrel needs to be maintained. Highly viscous materials are challenging and require high process temperatures and extrusion force. Processing high-viscous materials requires a longer printing time than low-viscous materials [40,41]. The materials exhibit thixotropic behavior where viscosity decreases with increasing temperature, but it is crucial to process materials with a low degradation temperature. For semi-solid materials, viscosity can be reduced by increasing the amount of solvent. However, the complete evaporation of the solvent needs to be ensured during the curing process to avoid toxicity to the patient population. In addition, the viscosity of the material extruded from the nozzle also needs to be studied. The high viscosity of the extruded material might affect the bonding between successive layers, and the extrusion of low-viscous materials might result in the deformation of shape due to the pressure exerted by the layers deposited above. The material extrusion rate from the nozzle also needs to be controlled since faster extrusion also results in the deformation of shape, and a slower extrusion rate results in the drying of material before reaching the build platform, affecting the product quality [27]. The optimum yield stress of the material inside the barrel prevents any premature flow of material, and consistent flow is maintained under pressure. The optimum yield stress also ensures the structural integrity of the 3D design without undergoing any deformation under pressure. Most of the material exhibits shear-thinning behavior, where viscosity decreases with increasing pressure, which is beneficial for the PAM process. Thus, processing temperature and pressure are essential in maintaining the optimum rheological properties for successful fabrication.

#### 2.1.3. Miscibility

The miscibility of formulation components alone and in the presence of a drug needs to be studied. Immiscible components are unsuitable for developing robust dosage forms and need to be avoided. Miscible formulation composition ensures the homogeneous distribution of drugs and polymers in the dosage form. Immiscible systems affect drug release profiles and eventually affect the oral bioavailability of the drug. For the solvent-aided process, solvent compatibility with formulation materials needs to be evaluated [42,43]. Immiscible compositions result in a non-homogeneous mixture and result in the non-uniform extrusion of material from the printer, affecting the quality, performance, and mechanical properties of the 3D-printed medication. In fact, immiscibility results in the non-homogeneous distribution of polymers in the case of modified medications, affecting the dosage form’s release profiles. A thorough preformulation study must be conducted to screen the drug, excipients, and suitable solvent system. Various characterization tools such as differential scanning calorimetry (DSC), Fourier transform infrared spectroscopy (FTIR), and scanning electron microscopy (SEM) can be employed to screen the miscibility of formulation components. Thus, miscibility is a critical factor influencing homogeneity, printability, mechanical properties, drug release profiles, and dosage form stability.

#### 2.1.4. Density of Material

The density of materials plays an important role in the segregation and non-homogeneous extrusion of material from the nozzle, affecting the quality of the dosage form. In the solvent-aided process, the particle size distribution of any undissolved materials needs to be monitored, which might result in the clogging of the nozzle, eventually interrupting the printing process. Screening materials with similar particle size distribution and density results in better product design and development. In other aspects, the printing of high-density materials requires the application of high pressure, and low-density materials will flow easily from the printer nozzle, which also influences the geometry of the 3D design [44]. The processing of high-density materials results in the high resolution of the dosage form, attributing to high melt viscosity and poor material spreading. Formulations composed of low-dense materials produce high-porosity objects, affecting mechanical properties and drug release profiles. In contrast, high-dense materials result in less porous objects, delaying drug release [45]. Thus, depending on the type of dosage form (immediate or modified release) being developed, materials can be screened based on density.

### 2.2. Critical Instrument Parameters

#### 2.2.1. Type of Piston

The instrumentation of PAM can accommodate different types of pistons, such as screw-driven pistons, pneumatic pistons, and hydraulic pistons. Screw-driven pistons move in rotational and linear motion, conveying the material toward the nozzle. The conveying material builds pressure at the nozzle and extrudes the material and is deposited on the build platform. By employing a screw type of piston, the entrapped air in the material can be removed, and also, the rotational motion helps in maintaining homogeneity. Screw speed dictates the flow rate of the material from the nozzle [46]. However, screw-driven pistons are not suitable for the processing of high-viscous materials, and the consistent dosing of material is challenging. In the case of pneumatic pistons, compressed air is applied onto the pistons to push the material from the nozzle. Compared with the screw-driven process, pneumatic pistons are more straightforward and can be operated at high speed. Pneumatic pistons are also capable of extruding highly viscous materials. However, the process is noisy, and maintaining a consistent air pressure is challenging, impacting the flow rate and quality of the medication. The other type of piston most commonly employed in the PAM process is the hydraulic piston, which utilizes pressurized liquids. Similar to the pneumatic process, this process is also suitable for processing highly viscous liquids, and pressure can be maintained consistently without affecting the flow rate of material from the nozzle [26]. It requires frequent maintenance to prevent the leakage of pressurized liquids, which makes the process expensive. Thus, each piston type is significant for processing materials. A suitable piston needs to be selected depending on the material properties.

#### 2.2.2. Nozzle Size

Nozzle size is another essential instrument component that plays a crucial role in successfully fabricating medications using the PAM additive manufacturing process. The nozzle size determines the resolution and quality of the printed dosage form. A smaller nozzle size results in a high-resolution printlet and increased printing time. It is also suitable for low-viscous materials. The printed dosage form’s surface appears smoother with a smaller nozzle. Printing highly viscous materials requires high pressure, and employing larger nozzles is beneficial. Employing larger nozzles reduces the printing time, but the resolution is compromised, resulting in a rough surface. Larger nozzles require less pressure for extruding materials and are suitable for highly viscous materials [47,48]. The size of the nozzle determines the layer thickness. In the case of any formulations with undissolved particles, employing larger nozzles is beneficial for successfully printing dosage forms. A wide range of nozzles ranging from 100 to 1000 microns in size are available. However, nozzle sizes ranging from 150 to 500 microns are most commonly utilized for pharmaceutical manufacturing. Thus, depending on the rheological properties of the formulations and the acceptability of dosage form resolution, the printer nozzle’s size must be selected.

#### 2.2.3. Cooling Fan

Within the instrumentation of PAM, the cooling fan is another component that needs to be considered as a critical aspect for the successful development of quality 3D-printed products. The cooling fan controls the temperature of the printing material and the printing environment. A high cooling fan speed is advantageous for preventing material overheating upon prolonged runs. At the same time, the cooling rate needs to be optimized to avoid the rapid drying of materials, which might affect bonding between layers [49]. The location of the fan also plays an important role in cooling printed objects. The uniform cooling of the object should be achieved, but non-uniform cooling affects the structural integrity and quality of the dosage form. A high cooling rate benefits low-viscous materials where rapid cooling prevents structural deformation. In the case of amorphous dispersions, a rapid cooling rate aids in the stability of the dosage forms. Thus, the cooling fan speed must be adjusted and optimized to print medications successfully.

#### 2.2.4. Build Platform

The build platform is essential and is a backbone for successfully printing dosage forms. The initial layer of the material extruded from the nozzle deposits and adheres to the build platform. Thus, the build platform supports the object being printed throughout the process. The build platform material needs to be compatible with the formulation components. Different build platform materials include glass, polyetherimide, polypropylene, polyethylene terephthalate, silicone, and polytetrafluoroethylene [27,39]. Towards the end of the printing process, the object needs to be ejected quickly without damaging the surface.

### 2.3. Critical Process Parameters

#### 2.3.1. Barrel and Nozzle Temperature

The process of PAM can be carried out under ambient conditions and by applying thermal energy depending on the physical state of the material being processed. In the solvent-aided process, where the material’s physical state is semi-solid, extrusion and printing can be carried out in ambient conditions by applying piston pressure. The processing of semi-solid material has to be carried out faster to avoid the evaporation of solvents, which might result in an increased viscosity of the processing material. Thermal energy might be needed to prevent nozzle blockage. However, the material’s thermal stability inside the barrel must be considered before applying heat. In the case of processing the powder material, the barrel needs to be heated to a temperature greater than the glass transition point of the processing material or greater than the melting point of the drug substance, depending on the intended physical state of the drug within the dosage form (amorphous or crystalline) [38,42]. To prevent the blockage of the nozzle and to maintain a consistent flow of the material, the nozzle has to be heated, and the temperature needs to be maintained consistently throughout the process. In the case of employing different barrel and nozzle temperatures, it should be ensured that no heat conduction occurs in either direction. The entire barrel has to be heated uniformly to prevent any variations in the melt viscosity, which might influence the flow and the quality of the dosage form. The temperature needs to be maintained entirely below the degradation point of the materials. Processing the powdered form of thermolabile materials remains challenging [34,36]. Processing powdered materials requires a longer heating time within the barrel to melt the formulation components. Thus, barrel and nozzle temperatures are essential in successfully fabricating medications.

#### 2.3.2. Printing Speed

Printing speed is a critical process parameter that determines the quality and mechanical integrity of the printed object. Printing speed, the viscosity of the processing material, and extrusion rate are interrelated and influence the quality, structural integrity, and successful printing of medications. A low printing speed requires a longer time to fabricate medicines. At slow speed, the deposited layers are dried up quickly and affect the bonding of newly deposited layers. A low printing speed might benefit low-viscous materials, providing enough residence time for the layers to become dried up and preventing structural deformation. A low printing speed results in a smooth surface of the extruding material from the nozzle, resulting in high-resolution medications. The piston pressure must be adjusted to match the extrusion rate with the printing speed at a low printing speed. A faster extrusion of material at low speed will result in the over-deposition of the material, affecting product quality [35,50]. High printing speeds can be employed for highly viscous materials, which do not cause any deformation of the 3D-printed structure. A high piston pressure must be applied at a high printing speed to increase the extrusion rate and match the deposition. Thus, printing speed plays a vital role in successfully printing the dosage form, and balance must be maintained between the printing speed and piston pressure.

#### 2.3.3. Piston Pressure

Depending on the physical state of the processing material and printing speed, the piston pressure needs to be applied and optimized. The processing of low-viscous materials requires low piston pressure, which maintains a consistent and uniform flow of material and results in the successful printing of medications [30,33]. The optimal piston pressure needs to be applied for low-viscous materials, and a low extrusion rate needs to be ensured, allowing sufficient time for the deposited layers to solidify and preventing the deformation of the printed material. Each type of piston can withstand different levels of pressure. The piston type needs to be optimized based on the amount of pressure being applied. The piston pressure has to be maintained below the maximum resistance limit to avoid any damage, resulting in the interruption of the process. The barrel temperature must be increased in a few situations to reduce the melt viscosity and execute the process at low piston pressures. In the case of high printing speed, a high piston pressure needs to be applied to maintain the extrusion rate of the material in line with the printing speed.

#### 2.3.4. Build Platform Temperature

During the additive manufacturing process, the build platform serves as a backbone, providing support for printed medication. The build platform temperature plays an important role in successfully printing dosage forms. The need to maintain the build platform temperature depends on the nature of the material being processed. Few materials adhere to the heated build platform, resulting in unsuccessful printing. Thus, the build platform should be heated, and the temperature needs to be optimized during the early stages of development. In a few instances, irrespective of adjusting the build platform temperature, the materials fail to adhere to the build platform, indicating that different construction materials should be investigated for the build platform. Maintaining a heated build platform also aids in the easy ejection of the printed medication, which deforms or affects the surface quality [37]. The optimal temperature needs to be maintained for successful processing. High build platform temperatures affect the viscosity of the initial later, thereby resulting in the loss of bonding between the platform and the printed medication. Similarly, a low platform temperature results in the poor bonding of the initial layer with the platform, resulting in the unsuccessful printing of dosage forms.

## 3. FDM 3D Printing

FDM 3D printing is a nozzle-based additive manufacturing technique that requires drug-loaded polymeric filaments as “ink” for printing medications. An FDM 3D printer typically consists of a gear roller, printer head, and build platform. FDM 3D printing begins with creating a 3D design, which is transferred as an stl file into the printer software, which controls the printer. The software conveys the information to the printer as G-code layer by layer until the fabrication is completed. FDM 3D printing involves heating and softening polymeric filaments within the printer head and depositing them on top of the build platform until the object of the desired size and geometry is formed [51,52]. The gear roller conveys the polymeric filament into the printer head, which serves as a piston for extruding the molten mass through a heated nozzle onto the build platform. A schematic representation of an FDM 3D printer is shown in Figure 2. Various process parameters involved in the FDM 3D printing process include printing speed, nozzle temperature, and build platform temperature.

The drug-loaded polymeric filaments required for FDM 3D printing can be prepared by impregnation or using a hot melt extrusion (HME) process. Impregnation involves the soaking of commercial filaments in solvent-based drug solutions, which results in low drug loading, and a careful screening of compatible solvents is needed to avoid any solvent-induced damage to the filaments [53,54]. Drying the filaments to remove any toxic solvents altogether is also challenging. HME is most widely investigated for manufacturing drug-loaded filaments suitable for the FDM process. HME is a solvent-free, single-step process that can also be coupled with an FDM 3D printing process for establishing a continuous manufacturing line [55,56]. Various critical material, instrumental, and process parameters that play crucial roles in successfully fabricating pharmaceutical medications by employing the FDM process are discussed below.

### 3.1. Critical Material Properties

#### 3.1.1. Mechanical Properties

The mechanical properties of filaments play an essential role in the successful printing of medications employing the FDM 3D printing process. Following the HME process, the filaments are collected as spools and loaded onto FDM 3D printers to fabricate dosage forms. The filaments should have enough flexibility to be rolled into spools, making handling easier. While feeding the filaments into the printer head, the filaments have to pass between the gear rollers, pushing the filaments continuously towards the melting zone in the heated nozzle. The gear rollers exert a certain mechanical pressure on the filaments during the process [57,58]. Brittle filaments become damaged between the gear rollers, and soft filaments become coiled inside the printer head, interrupting the process. Thus, a balance between brittleness and flexibility needs to be maintained to be suitable for the FDM process. As the gear rollers push the filaments towards the melting zone, the filaments also act as a piston for moving the molten material outside the nozzle. The mechanical properties of the filaments also become altered by any change in the filament diameter. Thus, ensuring the consistent extrusion of filaments is crucial. Any impact of the conveyor belt on the filament dimensions has to be controlled. An inconsistent material feed rate into the extruder barrel affects the fill volume and throughput, influencing the filament properties. Thus, the extrusion process has to be carefully optimized, and the automation of the process would eliminate human error [59]. To date, acceptance criteria for mechanical properties such as breaking force have yet to be established for the filaments to be suitable for the FDM 3D process since the mechanical properties greatly vary based on the filament composition.

#### 3.1.2. Surface Morphology

The surface morphology of filaments plays an important role in the successful printing of dosage forms. Filaments with rough surface morphology get stuck inside the printer head, interrupting fabrication. The surface morphology of filaments also influences the resolution of the printed medication. Thus, a smooth surface of filaments needs to be ensured to develop quality medications. The surface morphology of filaments is affected by the HME process. The extrusion of materials at low temperatures where the processing material is highly viscous would result in a rough surface of filaments [60,61]. The presence of any undissolved particles would also result in a rough surface. The presence of undissolved particles not only influences the resolution of medications but also results in the blockage of the printer nozzle. The resolution of the medications printed using rough surface filaments can be controlled by increasing the printing temperature and reducing the printing speed. Thus, the surface morphology of filaments needs to be controlled during the extrusion process.

#### 3.1.3. Filament Dimension

The dimensions of filaments are another critical material quality attribute that plays an important role in determining the suitability of filaments for the FDM 3D printing process. Polymeric chains undergo compression during the extrusion process due to the high mechanical stress exerted on the processing materials. Following the extrusion of filaments from the extruder die, the polymeric chains will relax, resulting in the swelling of the filament diameter, called “die swell” behavior [62,63]. Each type of FDM 3D printer accommodates different diameters of filaments. The die size connected to the extruder barrel must be selected based on the type of FDM 3D printer. Most commonly, FDM 3D printers accommodate filaments with diameters between 1.75 and 2.85 mm. Any filaments greater than the diameter the 3D printer can accommodate will not be suitable for processing. There is a need for advancement in the instrumentation of FDM 3D printers’ efficiency to accommodate different filament dimensions between gear rollers. During the process of HME, the die size needs to be selected based on the nature of polymeric materials. For materials with die swell behavior, the size of the die nozzle needs to be selected to be smaller than the gap between the gear rollers, ensuring the suitability and processability of filaments. Thus, filament dimensions are important and must be controlled during the HME process.

#### 3.1.4. Rheological and Thermal Properties

Similar to the process of PAM or SSE, the rheological and thermal properties of processing materials play an essential role in the successful fabrication of medications. The melt viscosity and flow of the material are critical properties that must be carefully monitored. Both the melt viscosity and flow of the material from the printer nozzle are temperature-dependent, which can be controlled by increasing or decreasing temperature. The degree of bonding between layers is higher for low-viscous materials than for high-viscous materials. High-viscous materials require a high amount of pressure to be exerted by the filament to push the molten material outside the printer nozzle, which might be challenging for brittle and soft filaments. The processing of highly viscous materials requires high extrusion and printing temperatures [64,65]. However, the processing temperature must be maintained below the degradation temperature of the drug substance, polymers, and other additive materials. In the case of high processing temperatures, the incorporation of plasticizers has to be studied to lower the melt viscosity and allow the glass transition temperature to process at low temperatures. For the printing of low-viscous materials, the temperature needs to be controlled to prevent the deformation of the object being fabricated. Compared with the PAM process, FDM 3D printing is unreliable for highly viscous materials and requires high processing temperatures to reduce the melt viscosity [66]. Thus, process temperature needs to be carefully optimized to control the rheological properties of materials for the successful fabrication of quality medications.

### 3.2. Critical Instrument Parameters

#### 3.2.1. Gear Rollers

Within the instrumentation of an FDM 3D printer, gear rollers are the key components that play a crucial role in successfully fabricating medications. Gear rollers are essential in feeding filaments into the printer head. Different gear rollers are available, such as normal, dual-gear, multi-gear, and tracked gear roller systems, as shown in Figure 3. The majority of commercial FDM 3D printers are equipped with dual-gear roller systems. Compared with dual and multi-gear roller systems, the normal and tracked roller systems exert a low level of mechanical shear on filaments, making brittle filaments suitable for the process [67,68]. The location of gear rollers on the instrument also plays an important role in successful printing. A greater distance between the nozzle and gear rollers exerts greater mechanical pressure on filaments, making it unsuitable for brittle materials. Having the flexibility to change different gear roller systems for the same instrument is beneficial for processing filaments with different mechanical properties.

#### 3.2.2. Nozzle Size

The nozzle size is important in determining the resolution and printing time. Within FDM 3D printers, the nozzle is the heating zone, which softens the filament and extrudes outside for deposition onto the build platform. With increasing nozzle size, the resolution of the printed material is lowered [69,70]. Conversely, the printing time can be reduced by increasing the nozzle size. Different nozzle sizes are available on the market, ranging from 0.1 to 1.0 mm. A nozzle size of less than 0.4 mm results in a longer printing time but delivers a high-resolution medication. Different construction nozzle materials, such as brass and stainless steel, are available on the market. Depending on the type of material being processed and the temperature required, the nozzle size and construction material need to be selected. For the processing of low-viscous materials, a smaller nozzle size is beneficial for the faster drying of the deposited layer and to prevent any deformation. For the processing of high-viscous materials, a nozzle with a larger diameter is recommended. Utilizing a larger nozzle to process viscous materials requires less mechanical pressure to be applied to filaments [71]. The thickness of each printed layer depends on the diameter of the nozzle. Thus, the nozzle size has to be chosen based on the quality of the printed medication.

#### 3.2.3. Cooling Fan and Build Platform

Following the extrusion of materials from the printer nozzle, the cooling fan and build platform are the two other key instrument components determining the successful fabrication of medications. Most of the FDM 3D printers available on the market do not have any control over adjusting the cooling fan’s speed. A faster cooling fan speed results in the rapid drying of the material before depositing it onto the build platform and affects the bonding and mechanical properties of the printlet. Similarly, a lower cooling rate delays the drying process of low-viscous materials, which might result in the deformation of the structure [72]. Thus, the optimum cooling rate must be maintained to fabricate medication successfully.

The build platform provides the foundation support to the printing object throughout the process. The loss of bonding between the bottom layer and the build platform interrupts the printing process. The material used to construct the build platform determines the success rate of the printing process. Only some polymers have superior adhesion properties with the build platform. Depending on the nature of the processing materials, the compatible platform needs to be identified [73]. Towards the end of the fabrication process, the object has to be quickly ejected without damaging the surface, which might influence the product’s performance. The commercial equipment should be flexible when switching between different build platforms, depending on the compatibility of processing materials. The platform construction material must be thermally stable to be suitable for the FDM 3D printing process.

### 3.3. Critical Process Parameters

#### 3.3.1. Printing Temperature

The printing temperature is a critical process parameter determining the successful fabrication and quality of medications. The process of FDM 3D printing requires the application of thermal energy to melt drug-loaded polymeric filaments. The process cannot be carried out at ambient temperatures and is impractical for thermolabile materials. Compared with the PAM or SSE process, the FDM process requires a higher temperature to melt filaments. The material is extruded from the nozzle using the gentle application of pressure by the unmelted filament in the printer head [74,75]. The filament inside the printer head might not be capable of applying high pressure attributed to its mechanical properties compared with the pressure applied by the pistons in the PAM or SSE process. The processing materials’ thermal properties must be studied, and the temperature must be maintained below the degradation temperature. Additionally, the nozzle temperature needs to be maintained below the melting point of the nozzle material of construction. The maximum temperature recommended by the equipment manufacturer has to be strictly followed. In the case of high-viscous formulations where low viscosities cannot be achieved below the degradation temperature, the incorporation of plasticizers has to be studied. Usually, the FDM process requires a temperature 10–20 °C greater than the HME temperature for printing dosage forms [76]. Thus, various aspects need to be considered when optimizing the nozzle temperature.

#### 3.3.2. Printing Speed

Printing speed is another critical process parameter that determines the mechanical integrity and quality of 3D-printed medications. Printing speed needs to be optimized based on various factors, such as the viscosity of the processing material and the binding property of the material. A higher printing speed for low-viscous materials will result in the deformation of printed dosage forms, affecting quality. Meanwhile, a faster printing speed for highly viscous materials will provide superior bonding and mechanical integrity for dosage forms [77,78]. However, slower printing speeds will result in a high resolution of 3D-printed medications.

#### 3.3.3. Build Platform Temperature

Similar to the build platform material, the temperature of the build platform is also an essential critical process parameter for the successful fabrication of quality medications. Higher build platform temperatures might result in the poor adhesion of printing materials with the platform due to the melting or softening of the initial support layer interrupting the printing process. Maintaining a low bed temperature also results in poor material bonding with the platform [79,80]. Thus, the build platform temperature needs to be carefully optimized to develop 3D medications successfully. Maintaining a higher build platform temperature might also result in the dehydration of deposited material, resulting in shrinkage and the loss of bonding between successive layers, affecting mechanical properties [81].

## 4. Coupling with HME

HME is a single-step continuous manufacturing process introduced earlier in the food, plastic, and rubber industries. Later, in the 1980s, the suitability of the HME process for developing amorphous solid dispersions of poorly water-soluble drug substances was investigated [82,83]. With the successful adaptation of the HME platform into the pharmaceutical industry for developing amorphous solid dispersions, its suitability for various other applications within the pharmaceutical sector was also studied [84,85,86]. Today, the HME process is most widely utilized in the pharmaceutical industry for developing amorphous solid dispersions and also for performing various types of granulations called twin-screw granulation. Since it is a solvent-free and single-step manufacturing process, HME was employed in various continuous manufacturing lines coupled with suitable downstream manufacturing equipment and process analytical technology (PAT) tools [87,88,89]. The instrumentation of HME includes a barrel enclosed with co-rotating or counter-rotating twin screws, a feeder, and a die connected at the discharge point. The physical blend of the formulation is fed into the heated extruder barrel where the material is exposed to thermal and mechanical shear [50,90,91]. The screws inside the barrel consist of various elements, such as conveying and kneading elements. The conveying elements have no mixing properties but convey the material between mixing zones. The mixing elements exert a high level of mechanical shear onto the processing material and influence distributive and dispersive mixing. The kneading elements can be configured into different levels of offset angles depending on the level of shear needed to be applied. With the increasing offset angle, the shear imparted onto the processing material will increase, and the conveying property will decrease. The processing material inside the barrel will be conveyed and pumped through the die connected at the discharge point [92,93,94,95,96]. During the granulation process and the extrusion of semi-solid pastes, the die connected at the discharge point can be removed. The critical process parameters include feed rate, barrel temperature, screw speed, torque, and screw configuration [97].

In recent years, researchers from academia and industries have worked closely to couple HME with additive manufacturing processes for developing patient-centric medications. HME is the primary process used for developing drug-loaded polymeric filaments used as stock material for fabricating 3D-printed medications using the FDM process. The filaments collected from the extrusion process can be transferred to compounding pharmacies where on-demand patient-centric medications can be fabricated [98,99,100]. To transform compounding pharmacies into digital pharmacies, the filaments’ shelf life must be established. In addition, HME can be paired with the FDM 3D printing process for developing a continuous manufacturing line. Various PAT tools can be coupled to monitor the homogeneous distribution of drug substances, the diameter of filaments, surface morphology, printing temperature, and build platform temperature [101,102].

Similarly, the process of PAM or SSE can also be coupled with the HME process. By coupling HME with the PAM or SSE process, pre-filled formulation cartridges can be prepared and shipped to compounding pharmacies, where the desired medication can be printed for different doses and release profiles depending on the patient’s needs [103,104]. Coupling with HME will also ensure and improve the homogeneous distribution of the drug within cartridges. In the case of a solvent-assisted process, cartridges can be supplied with powdered formulations, and semi-solid pastes can be prepared instantly by adding the required amount of solvent before printing. Along with a semi-continuous process, a continuous manufacturing line can be developed by coupling HME with PAM or SSE equipment. Various PAT tools can be used to monitor and control the critical process parameters and quality attributes of medications [35,105,106]. A detailed instrumentation of HME coupled with PAM or SSE and FDM platforms is shown in Figure 4. For more detailed information and an in-depth understanding of the continuous manufacturing process, readers are also suggested to refer to [107,108,109,110,111,112,113,114,115]. A detailed comparison of the HME, FDM, and PAM techniques is shown in Table 1. 

## 5. Quality by Design (QbD) Elements

QbD plays an important role in the successful development of drug products. A thorough understanding and control of critical material attributes (CMAs) and critical process parameters (CPPs) and their impact on the critical quality attributes (CQAs) of drug products will enable the development of safe, efficacious, and quality products. ICH guidelines Q8(R2) (Pharmaceutical Development) discuss the implementation of QbD elements for developing robust medications in detail [116,117]. Implementing these guidelines will give an in-depth understanding of the process and formulation parameters and build the product’s quality. The optimization of the process will ensure the reproducibility of medications. The main goal of QbD is to develop a reliable and reproducible process for delivering safe, quality, and efficacious medicines to the patient population. QbD involves a step-by-step implementation of various QbD elements discussed below. Implementing QbD for developing patient-centric medications by employing an additive manufacturing process is essential for ensuring consistency in product quality [118]. Much research is needed to understand CMAs, CPPs, and their impact on the CQAs of drug products.

### 5.1. Quality Targeted Product Profile (QTPP)

Any development activity begins with the drafting of the QTPP element. It serves as a reference guide for formulation scientists throughout the development phase. It provides developmental scientists with a blueprint for the characteristics of the drug product. Various elements of QTPP include dosage form, dosage design, route of administration, dosage strength, pharmacokinetics, stability, drug product quality attributes (appearance, assay, impurities, water content, content uniformity, dissolution, residual solvents, and microbial limits), container closure system, administration instructions, and alternative methods of administration.

### 5.2. CQAs

The CQA elements are listed within the QTPP section. However, the impact of process and material attributes on the CQAs of the drug product needs to be carefully monitored. Failure to maintain the CQAs of the drug product will affect the safety and efficacy of the product, resulting in potential risk to the patient population [119]. Throughout the shelf life, the CQAs of the product need to be preserved. Prior knowledge, experience, and literature review are important for identifying the CQAs during early-stage development.

### 5.3. CMAs

The CMAs of the drug substance and the excipients utilized in the design and development of drug products play an important role and influence CQAs. The identification of CMAs will help identify and optimize CPPs. For example, the degradation temperature of the drug substance can be considered as a CMA, due to the thermal processing conditions. Based on the CMA of the drug substance, the process temperature can be identified as a CPP and must be maintained below the degradation temperature of the drug substance [120].

### 5.4. CPPs

The identification of CPPs requires previous knowledge of and experience with the process and the identification of CMAs impacted by process parameters, which will eventually affect the CQAs of the finished drug product [121].

### 5.5. Risk Assessment

Risk assessment is another critical aspect of building quality products. Once the CMAs, CPPs, and CQAs are identified, the effect of CMAs and CPPs on the CQAs of the drug product needs to be identified, and the level of risk needs to be determined [122,123]. The initial risk assessment involves the evaluation of risk associated with drug substance attributes (solid state form, particle size, solubility, moisture content, residual solvents, process impurities, chemical stability, and flow properties), formulation variables (drug substance particle size, excipients, flow properties), and process variables (the effect of each unit operation process variable on the CQAs needs to be assessed), and their effects on the CQAs of the drug product need to be assessed. Implementing risk assessment tools such as fish-bone diagrams or failure mode effect analysis (FMEA) will aid in better risk assessment tracking [105]. In addition, by employing the design of experiments (DoE), the effect of CMAs and CPPs on the CQAs of drug products can be studied in detail, and this helps reduce risk severity by operating the process within the design space.

### 5.6. Control Strategy and Continuous Improvement

Based on the knowledge gained from product and process development, a control strategy can be defined to ensure the consistent delivery of product quality. A range or limit can be established for each aspect and must be strictly followed to develop a quality product. The various control strategies are discussed below.
CQAs: CQAs need to be within the established limits to ensure the quality of the product.CMAs: The drug substance and the excipient should comply with the specification and satisfy the pre-determined requirements to eliminate any associated risk.CPPs: The process needs to be operated within the qualified and validated ranges to prevent any unwanted effects on the CQAs of drug products.In-process controls: Throughout the manufacturing process, the critical process parameters must be monitored continuously to ensure they stay within the established range and the intermediate product meets the pre-established specification, directly influencing the CQAs of the final product.Specifications: An acceptance criterion needs to be established within the specification to ensure product quality.PAT: The implementation of PAT tools will help monitor CMAs and CPPs, which are directly associated with the CQAs of drug products.Real-time release testing (RTRT): RTRT involves the in-line testing of the intermediate and finished drug product. Implementing RTRT will increase productivity and reduce costs. For example, the uniformity of the drug within the formulation blend can be monitored by mounting a near-infrared (NIR) probe. This process will ensure the development of a quality and safe medication. Any discrepancies can be controlled immediately without being realized towards the end of the manufacturing during end-product testing.

Even after commercialization, the continuous monitoring and improvement of product quality are needed. Trends in the manufacturing process need to be continuously monitored and corrected as needed. Quality metrics need to be implemented to track yield and any defects. Any discrepancies must be resolved by proper measures utilizing corrective and preventive action (CAPA). Feedback from the manufacturing and end-product users needs to be considered to find any improvement areas. Innovative technologies can be implemented to simplify the process and reduce the probability of incidents. Various elements of QbD are listed in Figure 5.

## 6. Future Perspectives and Expert Opinion

In recent years, additive manufacturing has been the most widely explored method for developing patient-centric personalized medications. Much research is ongoing in academia and industries, exploring the suitability and capability of various additive manufacturing techniques for developing pharmaceutical medications [124,125]. In 2021, the FDA approved an investigational new drug (IND) application for conducting first-in-human (FIH) clinical trials for rheumatoid arthritis using a 3D-printed medication manufactured by Triastek, a Chinese pharmaceutical company specialized in developing 3D-printed dosage forms [126,127]. This shows pharmaceutical companies’ interest and future focus and the encouragement of regulatory agencies to explore and adapt novel manufacturing technologies for delivering safe and efficacious medications to the patient population. Additive manufacturing has been successfully implemented for developing various complex medical devices, and even regulatory agencies such as the FDA have released specific guidelines for industries. Employing additive manufacturing will address various limitations associated with the current manufacturing technologies. The limitations related to tableting and encapsulation, such as poor compressibility, poor flow, capping, lamination, and high friability, can be avoided, and developmental timelines and costs can be reduced drastically [128,129]. The scaleup of the additive manufacturing process can be achieved by increasing the printing speed and the number of printer heads on the build platform. Medications can easily be fabricated in clinical sites for different dosage strengths without initiating reformulation activities. Usually, any formulation composition or manufacturing change requires establishing the bioequivalence between two formulations, which is expensive and more time-consuming. Pharmaceutical industries will not be willing to change formulations to ensure the product is launched onto the market. Still, they will face high-level challenges during scaleup and commercial manufacturing [130]. By employing additive manufacturing, any change in dose can be easily adjusted without changing the formulation composition, and release profiles can be adjusted by studying different infill densities and patterns. This will save a lot of investment and time for the pharmaceutical industry.

In the future, the nozzle-based additive manufacturing techniques discussed in the current review article can transform into a single-step continuous manufacturing line by coupling them with the HME process. Various PAT tools can be employed to monitor critical process parameters and quality attributes. Any discrepancy in the process can be monitored and controlled, thereby developing a safe, efficacious, and quality product for the patient population. In addition, both PAM and FDM platforms can be established within compounding pharmacies in hospitals, and the material required for the fabrication process, such as pre-filled cartridges for PAM and drug-loaded polymeric filaments for the FDM process, can be supplied by pharmaceutical companies. Thus, compounding pharmacies can be transformed into digital pharmacies to fabricate on-demand medications. Pediatric medications with different geometries, colors, and flavors can be fabricated to attract children and ensure that they adhere to taking prescription medications. Complex medications such as polypills can be manufactured easily without challenging multilayer traditional compression processes [131,132,133]. Overall, the additive manufacturing process for developing pharmaceutical medications answers various unmet needs of the pharmaceutical industry and the patient population.

A detailed SWOT (strength, weakness, opportunities, and threats) analysis critical for additive manufacturing is shown in Figure 6. As discussed earlier, additive manufacturing benefits both industries and the patient population. When compared with traditional manufacturing platforms, additive manufacturing provides the advantage of personalized medication and enables the opportunity for developing complex medications right in the clinic under the supervision of physicians. In today’s world, no manufacturing process is free of drawbacks or limitations. Similar to traditional manufacturing techniques, additive manufacturing also has a few limitations, but they can be addressed in the future with the advancement in technology. The limitations of the additive manufacturing platform seem to be less concerning due to the significant advantages it can provide to the pharmaceutical sector. The major limitations of additive manufacturing include the availability of suitable materials, throughput of the process, and the lack of specific regulatory guidelines for pharmaceuticals. All the above-mentioned limitations, if they remain unaddressed, might eventually result in higher product costs. There is a need for excipient manufacturers to introduce novel, cost-effective, and safe materials suitable for the additive manufacturing process. Excipients should majorly address the existing limitations of thermal and rheological aspects. Excipient manufacturers should work closely with the pharmaceutical industry to design and develop excipients that meet the industrial and regulatory requirements. The current benchtop capacity of the instruments that are available on the market can be modified according to the good manufacturing practice (GMP) guidelines and can be utilized for developing prototypes, for the on-demand manufacturing of complex emergency medications, and for manufacturing clinical supplies in clinical sites. Meanwhile, knowledge from various industrial sectors, such as automobiles where additive manufacturing is being employed on a large scale for developing complex components, can be obtained and utilized as a starting point for manufacturing pharmaceutical-grade equipment to meet commercial needs. The recent initiation of clinical trials using the medications manufactured using the additive manufacturing process shows the interest and encouragement of regulatory authorities. To date, no specific guidelines for developing pharmaceutical medications using additive manufacturing techniques are available, which might be another limiting factor that prevents the pharmaceutical industry from implementing the additive manufacturing process. Though the first 3D-printed medication was approved in 2015, it has been almost a decade with no additional products being launched onto the market. All the above-mentioned limitations might be major factors that are making the pharmaceutical industry step back from transforming manufacturing technologies into additive manufacturing technologies. In addition, Artificial Intelligence (AI) and Machine Learning (ML) can be implemented for the rapid and effective development of pharmaceutical medications using additive manufacturing platforms.

## 7. Conclusions

In recent years, additive manufacturing has been widely explored for developing patient-centric medications. The complex medicines needed on demand for the patient population can be fabricated in emergencies. The performance of medicines can be altered by modifying the infill densities, infill pattern, and geometry of the 3D design without undertaking reformulation activities. Adapting the additive manufacturing process will provide advanced capabilities for the pharmaceutical industry and address various unmet needs of traditional manufacturing platforms. The additive manufacturing platforms PAM and FDM can be implemented in compounding pharmacies or coupled with the HME process to develop a continuous manufacturing process. The quality, safety, and efficacy of medications can be ensured by employing various PAT tools. Still, a lot of research needs to be conducted to understand the effects of various factors on the quality of medications before they are introduced into industries for commercial manufacturing. Novel next-generation materials must be developed to be compatible with thermal processing conditions. An advancement in instrumentation is also needed to comply with good manufacturing practice (GMP) standards, be compatible with the processing materials, and support commercial batch sizes. The successful implementation of additive manufacturing platforms and the launch of various commercial products will revolutionize the entire pharmaceutical history.

## Figures and Tables

**Figure 1 pharmaceutics-17-00014-f001:**
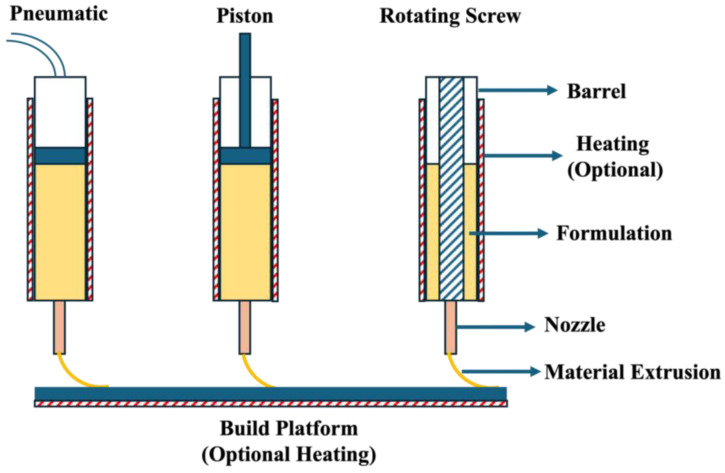
Detailed PAM instrumentation with different piston types.

**Figure 2 pharmaceutics-17-00014-f002:**
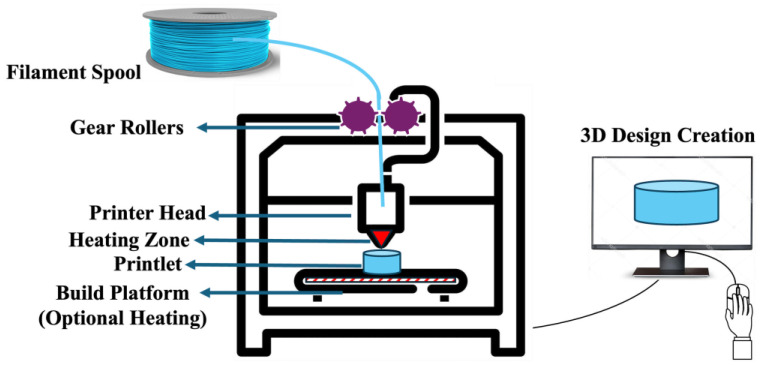
Detailed instrumentation of FDM platform.

**Figure 3 pharmaceutics-17-00014-f003:**
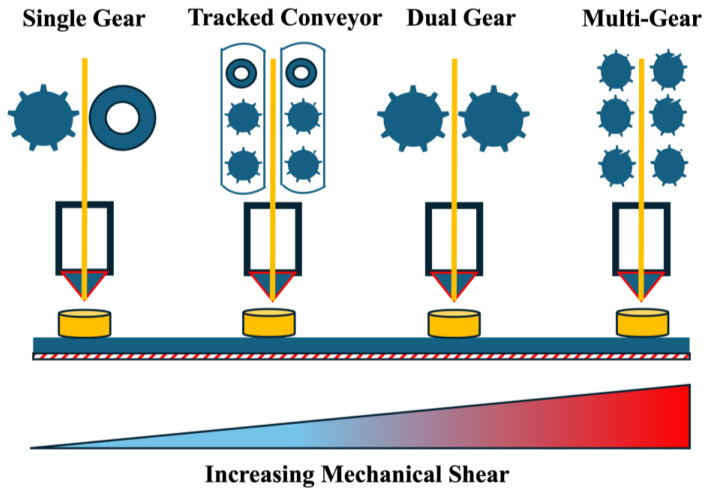
Different types of gear rollers available for FDM instruments and their level of mechanical shear.

**Figure 4 pharmaceutics-17-00014-f004:**
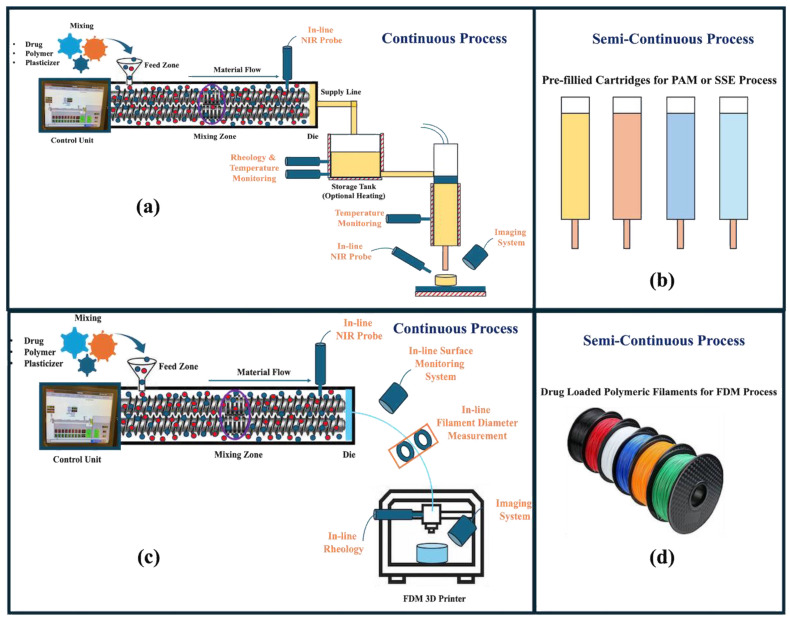
(**a**) Continuous manufacturing line for HME and PAM process; (**b**) pre-filled cartridges for shipping to compounding pharmacies; (**c**) continuous manufacturing line coupling HME and FDM process; (**d**) drug-loaded polymeric filaments for shipping to compounding pharmacies.

**Figure 5 pharmaceutics-17-00014-f005:**
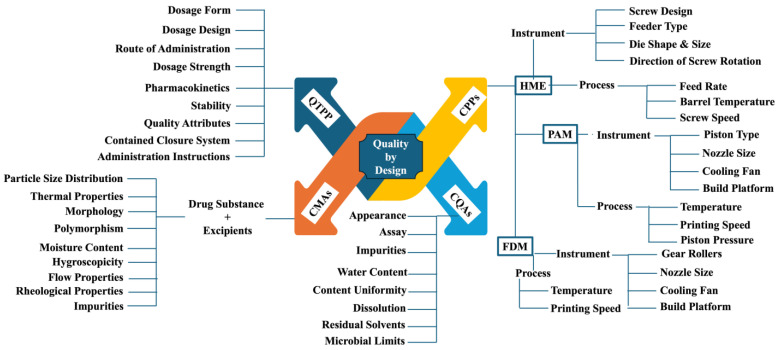
A detailed list of QbD elements critical for the successful development of pharmaceutical medications.

**Figure 6 pharmaceutics-17-00014-f006:**
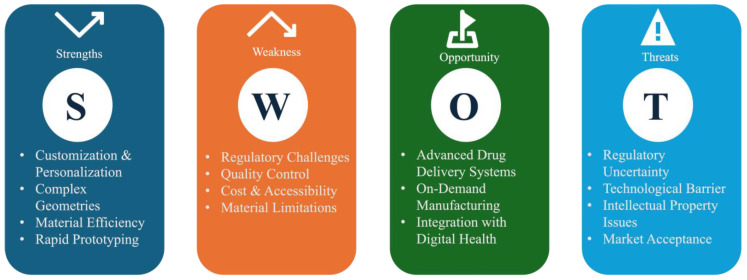
SWOT analysis of additive manufacturing for successful development of pharmaceutical medications.

**Table 1 pharmaceutics-17-00014-t001:** Detailed comparison of hot melt extrusion, fused deposition modeling, and pressure-assisted microsyringe techniques.

Process	Pros	Cons
HME	Single-step process Solvent-free Versatility for various dosage forms	Thermal process Not suitable for thermolabile materials High equipment and operational costs Limited polymer choices
FDM	Customization and personalization Flexibility in formulations Cost-effective for small batches Rapid prototyping On-demand manufacturing	High processing temperature Material constraints Resolution and precision issues Regulatory challenges Scalability
PAM	High precision and resolution Low processing temperature Suitable for thermolabile materials On-demand manufacturing	High equipment and operational costs Material limitations Production speed Regulatory challenges

## Data Availability

No new data were created or analyzed in this study. Data sharing is not applicable to this article.

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
