# Peer review of "Enhancing Patient-Centric Drug Development: Coupling Hot Melt Extrusion with Fused Deposition Modeling and Pressure-Assisted Microsyringe Additive Manufacturing Platforms with Quality by Design"

_pharmaceutics, 2024, doi:10.3390/pharmaceutics17010014_

Round 1
Reviewer 1 Report
Comments and Suggestions for Authors
The manuscript is a comprehensive review exploring the application of Hot Melt Extrusion (HME) and Nozzle-Based Additive Manufacturing Techniques such as Pressure-Assisted Microsyringe (PAM) and Fused Deposition Modeling (FDM) for designing and developing pharmaceutical medications. It emphasizes the increasing demand for patient-centric and personalized medications that traditional manufacturing methods cannot adequately address.
In general the manuscript is well presented but there are some areas that the authors may wish to address prior to acceptance for publication.
The title could be revised to emphasise the specificity of AM techniques (PAM, FDM) and the coupling with HME. Including references to continuous manufacturing or patient-centric design would better align it with the manuscript's forward-looking perspectives and not necessarily drug design and deveolpment as the review would need to be expanded to talk about additional formulation aspects.
The paper does not adequately address why there has been no FDA-approved 3D printed medication since Spritam. While it mentions the approval of Spritam as a significant milestone and highlights the potential of additive manufacturing (AM) techniques, it does not explore the reasons for the lack of subsequent approvals in sufficient depth. This approval would have been around 2015/2016 and even with the advance of additive manufacturing technologies and increased research in the area of additive manufacturing there has been little progess in the area. A discussion of this would be most welcome and could include dscussions around regulatory hurdles, cost benefit and scalability.
While it in no way detracts from the overall readability of the manuscript, some care should be taken with regards to some typographical and grammatical errors.(which are to be expected on a review of this length). Another careful review and edit by the authors is warranted prior to publication.
Line 56 - The product in question Spitram® and is manufactured using the Zipdose® technology platform
Line 277 - "more significantly" seems out of place. It probably should read "heated to a temperature 'greater than' or 'above' the glass transition"
Line 491 - Should probably read "Lots...."
Author Response
- The title could be revised to emphasise the specificity of AM techniques (PAM, FDM) and the coupling with HME. Including references to continuous manufacturing or patient-centric design would better align it with the manuscript's forward-looking perspectives and not necessarily drug design and deveolpment as the review would need to be expanded to talk about additional formulation aspects.
Response: Authors thank the Reviewer for the valuable inputs. As suggested, we have updated the title of article as shown below. Also, we have included more reference for the readers to get detailed and in-depth understanding of the continuous manufacturing process. The primary focus of the current review article is to discuss more about the impact of critical instrument, material and formulation parameters on the critical quality attributes of the medications. The authors are currently working to come up with another review article which will mainly focus on the formulation development aspects which is not covered in this article.
“Enhancing Patient-Centric Drug Development: Coupling Hot Melt Extrusion with FDM and PAM Additive Manufacturing Platforms with Quality-by-Design”
- The paper does not adequately address why there has been no FDA-approved 3D printed medication since Spritam. While it mentions the approval of Spritam as a significant milestone and highlights the potential of additive manufacturing (AM) techniques, it does not explore the reasons for the lack of subsequent approvals in sufficient depth. This approval would have been around 2015/2016 and even with the advance of additive manufacturing technologies and increased research in the area of additive manufacturing there has been little progess in the area. A discussion of this would be most welcome and could include dscussions around regulatory hurdles, cost benefit and scalability.
Response: Authors thank the Reviewer for the suggestion. The authors have updated the future perspective section of the manuscript discussing a few limitations of the additive manufacturing platform and why there have no additional products launched into the market.
“As discussed earlier, additive manufacturing benefits both the industries and patient population. When compared with traditional manufacturing platforms, the additive manufacturing provides the advantage of personalized medication and enables the opportunity for developing complex medications right in the clinic under the supervision of physicians. In today’s world, no manufacturing process is free of drawbacks or limitations. Similar to traditional manufacturing techniques, the additive manufacturing also has few limitations but can be addressed in the future with the advancement in the technology. The limitations of the additive manufacturing platform seem to be less concerned due to the significant amount of advantage it can add to the pharmaceutical sector. The major limitations of additive manufacturing include availability of suitable materials, throughput of the process, and lack of specific regulatory guidelines for pharmaceuticals. All the above-mentioned limitations if remains unaddressed might eventually result in higher product cost. There is a need for the excipient manufacturers to introduce novel, cost effective and safe materials suitable for additive manufacturing process. The excipients should majorly address the existing limitations of thermal and rheological aspects. The excipient manufacturers should work closely with the pharmaceutical industries to design and develop excipients which meets the industrial and regulatory requirements. The current benchtop capacity of the instruments that are available in the market can be modified according to the Good Manufacturing Practice (GMP) guidelines and can be utilized for developing prototypes, for on-demand manufacturing of complex emergency medications and for manufacturing clinical supplies at the clinical site. Meanwhile, knowledge from various industrial sectors such as automobiles where additive manufacturing is being employed for in large scale for developing complex components can be obtained and utilized as a starting point for manufacturing pharmaceutical grade equipment’s to meet the commercial needs. The recent initiation of the clinical trials using the medications manufactured by additive manufacturing process shows the interest and encouragement of the regulatory authorities. To date no specific guidelines for developing pharmaceutical medications by additive manufacturing techniques are available which might be another limiting factor for pharmaceutical industries from being implementing the additive manufacturing process. Though the first 3D pritnted medications is approved in 2015, it’s been almost a decade with no additional products being launched into the market. All the above mentioned limitations might be the major factors that are making the pharmaceutical industries to step back from being transforming the manufacturing technologies towards additive manufacturing.”
- While it in no way detracts from the overall readability of the manuscript, some care should be taken with regards to some typographical and grammatical errors.(which are to be expected on a review of this length). Another careful review and edit by the authors is warranted prior to publication.
Response: Authors thank the reviewer for the recommendation. As suggested, the entire manuscript has been reviewed and correct for typos and minor grammatical mistakes.
- Line 56 - The product in question Spitram® and is manufactured using the Zipdose® technology platform
Response: Authors thank the Reviewer for the comment. Unfortunately, the authors did not completely understand the suggestion provided by the Reviewer. However, we have corrected the sentence for the trade symbols.
- Line 277 - "more significantly" seems out of place. It probably should read "heated to a temperature 'greater than'or 'above' the glass transition"
Response: Authors thank the Reviewer for identifying the mistake. We have correct the statement as suggested.
Line 491 - Should probably read "Lots...."
Response: Authors thank the reviewer for the comment. The entire section has been revised.

Reviewer 2 Report
Comments and Suggestions for Authors
This review comprehensively explores the role of nozzle-based additive manufacturing techniques like pressure-assisted microsyringe and fused deposition modelling in developing patient-centric, on-demand medications, emphasizing critical process parameters, continuous manufacturing, and quality-by-design approaches. My comments are as follows:
1. Add a statement in the paragraph, about how this work is different than already published work on 3D printing. https://doi.org/10.1016/j.carbpol.2020.116519 etc.
2. Please provide an example of a research study in this field in tabular summarization.
3. Discuss the key factor and its effect with a suitable published research study. A detailed of the types of pharmaceutical-grade materials compatible with PAM and FDM should be there.
4. What regulatory conferences are there for HME Nozzle Based Additive Manufacturing Techniques
5. The review lacks a comparative analysis of nozzle-based additive manufacturing techniques with other additive manufacturing methods (e.g., binder jetting, and stereolithography) to highlight their relative advantages and limitations. Add a limitation, especially material suitability and cost.
6. Future perspective as far as technology is concerned. Refer (https://doi.org/10.23919/ACC50511.2021.9483241)
Author Response
- Add a statement in the paragraph, about how this work is different than already published work on 3D printing. https://doi.org/10.1016/j.carbpol.2020.116519
Response: Authors thank the Reviewer for the suggestion. The article https://doi.org/10.1016/j.carbpol.2020.116519 is a research manuscript and is focused on the development of gastri-floating tablets manufactured by FDM 3D printing technology. The primary focus of the current review article is to discuss more about the impact of critical instrument, material and formulation parameters on the critical quality attributes of the medications. In additional the manuscript has also discussed the feasibility of continuous manufacturing coupling with hot melt extrusion process along with quality by design approach. As suggested the novelty of the review article is highlighted in the introduction section of the manuscript.
- Please provide an example of a research study in this field in tabular summarization. Discuss the key factor and its effect with a suitable published research study. A detailed of the types of pharmaceutical-grade materials compatible with PAM and FDM should be there.
Response: Authors thank the Reviewer for the suggestion. Since the main focus of the current review article is to discuss the critical factors affecting the quality attributes of the dosage form and to discuss the feasibility of continuous manufacturing by coupling with hot melt extrusion, the case studies were not covered in this part of the article. The part II of this article will discuss more about the formulation development and case studied in detail along with limitation.
- What regulatory conferences are there for HME Nozzle Based Additive Manufacturing Techniques
Response: Authors thank the Reviewer for the comment. To date no regulatory guidelines have been published specifically for developing pharmaceutical medications by additive manufacturing platform. The recent initiation of the clinical trials using the drug product manufactured by additive manufacturing process shows the interest of the regulatory authorities. The information has been covered in the future perspective section of the article.
- The review lacks a comparative analysis of nozzle-based additive manufacturing techniques with other additive manufacturing methods (e.g., binder jetting, and stereolithography) to highlight their relative advantages and limitations. Add a limitation, especially material suitability and cost.
Response: Authors thank the Reviewer for the suggestion. Since the main focus of the article is to discuss about FDM and PAM technologies, the other techniques were not discussed. The limitations of the additive manufacturing for developing pharmaceutical medications has been updated and discussed in the future perspective section.
- Future perspective as far as technology is concerned. Refer (https://doi.org/10.23919/ACC50511.2021.9483241)
Response: Authors thank the Reviewer for the suggestion. The future perspective section has been updated discussing the current status and limitations of additive manufacturing platform for developing pharmaceutical medications.

Reviewer 3 Report
Comments and Suggestions for Authors
In this work the authors reviewed PAM and FDM technologies with possible coupling with HME. The paper is on a topic of importance and will be of interest to others working in the field. I recommend publication with minor changes.
· In PAM, consistent dosing in the barrel is critical, especially for the rotating screw setup. I recommend the authors adding this part in the MS.
· I suggest that the authors providing a table to compare the advantages and disadvantages of PAM, FDM and HME. What are the limitations of PAM and FDM?
· Modeling/simulations have been done for these processes. I suggest that the authors add the comments on this part.
Author Response
- In PAM, consistent dosing in the barrel is critical, especially for the rotating screw setup. I recommend the authors adding this part in the MS.
Response: Authors thank the Reviewer for the input. As suggested, the information has been updated in Section 2.2.1.
- I suggest that the authors providing a table to compare the advantages and disadvantages of PAM, FDM and HME. What are the limitations of PAM and FDM?
Response: Authors thank the Reviewer for the suggestion. A table with detailed comparison of HME, FDM and PAM techniques is added to the manuscript.
- Modeling/simulations have been done for these processes. I suggest that the authors add the comments on this part.
Response: Authors thank the Reviewer for the input. As suggested, the information pertaining to models, AI and ML has been added to section 6 of the manuscript.

Round 2
Reviewer 2 Report
Comments and Suggestions for Authors
Author incorporated all corrections as suggested. Accept the manuscript in present form.